# Beyond Fully-Connected Layers with Quaternions: Parameterization of Hypercomplex Multiplications with $1/n$ Parameters

**Aston Zhang**[†], **Yi Tay**[‡,*] **Shuai Zhang**[◇]**, Alvin Chan**[◁]
**Anh Tuan Luu**[◁,○]**, Siu Cheung Hui**[◁]**, Jie Fu**[●]
[†]Amazon Web Services AI
[‡]Google Research
[◇]ETH Zürich
[◁]NTU, Singapore
[○]VinAI
[●]Mila, Université de Montréal
az@astonzhang.com

## Abstract

Recent works have demonstrated reasonable success of representation learning in hypercomplex space. Specifically, "fully-connected layers with quaternions" (quaternions are 4D hypercomplex numbers), which replace real-valued matrix multiplications in fully-connected layers with Hamilton products of quaternions, both enjoy parameter savings with only $1/4$ learnable parameters and achieve comparable performance in various applications. However, one key caveat is that hypercomplex space only exists at very few predefined dimensions (4D, 8D, and 16D). This restricts the flexibility of models that leverage hypercomplex multiplications. To this end, we propose parameterizing hypercomplex multiplications, allowing models to learn multiplication rules from data regardless of whether such rules are predefined. As a result, our method not only subsumes the Hamilton product, but also learns to operate on any arbitrary $n$D hypercomplex space, providing more architectural flexibility using arbitrarily $1/n$ learnable parameters compared with the fully-connected layer counterpart. Experiments of applications to the LSTM and transformer models on natural language inference, machine translation, text style transfer, and subject verb agreement demonstrate architectural flexibility and effectiveness of the proposed approach.

## 1 Introduction

A quaternion is a 4D hypercomplex number with one real component and three imaginary components. The Hamilton product is the hypercomplex multiplication of two quaternions. Recent works in quaternion space and Hamilton products have demonstrated reasonable success (Parcollet et al., 2018b; 2019; Tay et al., 2019). Notably, the Hamilton product enjoys a parameter saving with $1/4$ learnable parameters as compared with the real-valued matrix multiplication. It also enables effective representation learning by modeling interactions between real and imaginary components.

One of the attractive properties of quaternion models is its high applicability and universal usefulness to one of the most ubiquitous layers in deep learning, i.e., the fully-connected (or feedforward) layer. Specifically, "fully-connected layers with quaternions" replace real-valued matrix multiplications in fully-connected layers with Hamilton products of quaternions, enjoying parameter savings with only $1/4$ learnable parameters and achieving comparable performance with their fully-connected layer counterparts (Parcollet et al., 2018b; 2019; Tay et al., 2019).

The fully-connected layer is one of the most dominant components in existing deep learning literature (Goodfellow et al., 2016; Zhang et al., 2020). Its pervasiveness cannot be understated, given its

---

[*]Work was done at NTU.

centrality to many core building blocks in neural network research. Given widespread adoptions of fully-connected layers, e.g., within LSTM networks (Hochreiter & Schmidhuber, 1997) and transformer models (Vaswani et al., 2017), having flexibility to balance between parameter savings and effectiveness could be extremely useful to many real-world applications.

Unfortunately, hypercomplex space only exists at 4D (quaternions), 8D (octonions), and 16D (sedenions), which generalizes the 2D complex space (Rishiyur, 2006). Moreover, custom operators are required at each hypercomplex dimensionality. For instance, the Hamilton product is the hypercomplex multiplication in 4D hypercomplex space. Thus, no operator in such predefined hypercomplex space is suitable for applications that prefer reducing parameters to $1/n$, where $n \neq 4, 8, 16$.

In view of the architectural limitation due to the very few choices of those existing hypercomplex space, we propose parameterization of hypercomplex multiplications, i.e., learning the real and imaginary component interactions from data in a differentiable fashion. Essentially, our method can operate on an arbitrary $n$D hypercomplex space, aside from subsuming those predefined hypercomplex multiplication rules, facilitating using up to *arbitrarily* $1/n$ learnable parameters while maintaining expressiveness. In practice, the hyperparameter $n$ can be flexibly specified or tuned by users based on applications.

Concretely, our prime contribution is a new module that parameterizes and generalizes the hypercomplex multiplication by learning the real and imaginary component interactions, i.e., multiplication rules, from data. Our method, which we call the parameterized hypercomplex multiplication layer, is characterized by a sum of Kronecker products that generalize the vector outer products to higher dimensions in real space. To demonstrate applicability, we equip two well-established models (the LSTM and transformer) with our proposed method. We conduct extensive experiments on different tasks, i.e., natural language inference for LSTM networks and machine translation for transformer models. Additionally, we perform further experiments on text style transfer and subject verb agreement tasks. All in all, our method has demonstrated architectural flexibility through different experimental settings, where it generally can use a fraction of the learnable parameters with minimal degradation or slight improvement in performance.

The overall contributions of this work are summarized as follows:

- We propose a new parameterization of hypercomplex multiplications: the parameterized hypercomplex multiplication (PHM) layer. This layer has $1/n$ learnable parameters compared with the fully-connected layer counterpart, where $n$ can be flexibly specified by users. The key idea behind PHM layers is to learn the interactions between real and imaginary components, i.e., multiplication rules, from data using a sum of Kronecker products.

- We demonstrate the applicability of the PHM layers by leveraging them in two dominant neural architectures: the LSTM and transformer models.

- We empirically show architectural flexibility and effectiveness of PHM layers by conducting extensive experiments on five natural language inference tasks, seven machine translation datasets, together with text style transfer and subject verb agreement tasks.

## 2 BACKGROUND ON QUATERNIONS AND HAMILTON PRODUCTS

We begin by introducing the background for the rest of the paper. Concretely, we describe quaternion algebra along with Hamilton products, which is at the heart of our proposed approach.

**Quaternion** A quaternion $Q \in \mathbb{H}$ is a hypercomplex number with one real component and three imaginary components as follows:

$$Q = Q_r + Q_x\mathbf{i} + Q_y\mathbf{j} + Q_z\mathbf{k}, \tag{2.1}$$

whereby $\mathbf{ijk} = \mathbf{i}^2 = \mathbf{j}^2 = \mathbf{k}^2 = -1$. In (2.1), noncommutative multiplication rules hold: $\mathbf{ij} = \mathbf{k}, \mathbf{jk} = \mathbf{i}, \mathbf{ki} = \mathbf{j}, \mathbf{ji} = -\mathbf{k}, \mathbf{kj} = -\mathbf{i}, \mathbf{ik} = -\mathbf{j}$. Here, $Q_r$ is the real component, $Q_x, Q_y, Q_z$ are real numbers that represent the imaginary components of the quaternion $Q$.

**Addition** The addition of two quaternions is defined as

$$Q + P = Q_r + P_r + (Q_x + P_x)\mathbf{i} + (Q_y + P_y)\mathbf{j} + (Q_z + P_z)\mathbf{k},$$

where $Q$ and $P$ with subscripts denote the real and imaginary components of quaternions $Q$ and $P$.

**Scalar Multiplication**    Any scalar $\alpha$ multiplies across all the components:

$$\alpha Q = \alpha Q_r + \alpha Q_x \mathbf{i} + \alpha Q_y \mathbf{j} + \alpha Q_z \mathbf{k}.$$

**Hamilton Product**    The Hamilton product, which represents the multiplication of two quaternions $Q$ and $P$, is defined as

$$Q \otimes P = (Q_r P_r - Q_x P_x - Q_y P_y - Q_z P_z) + (Q_x P_r + Q_r P_x - Q_z P_y + Q_y P_z)\,\mathbf{i}$$
$$+ (Q_y P_r + Q_z P_x + Q_r P_y - Q_x P_z)\,\mathbf{j} + (Q_z P_r - Q_y P_x + Q_x P_y + Q_r P_z)\,\mathbf{k}. \quad (2.2)$$

The multiplication rule in (2.2) forges interactions between real and imaginary components of $Q$ and $P$. The benefits of Hamilton products have been demonstrated in recent works where the matrix multiplication in fully-connected layers is replaced with the Hamilton product: this reduces 75% parameters with comparable performance (Parcollet et al., 2018b; 2019; Tay et al., 2019).

# 3    PARAMETERIZATION OF HYPERCOMPLEX MULTIPLICATIONS

The following introduces our proposed parameterized hypercomplex multiplication layer and elaborates on how it parameterizes and generalizes multiplications in hypercomplex space, such as subsuming the multiplication rules of Hamilton products in (2.2).

## 3.1    FULLY-CONNECTED (FC) LAYERS

Before we delve into our proposed method, recall the fully-connected (FC) layer that transforms an input $\mathbf{x} \in \mathbb{R}^d$ into an output $\mathbf{y} \in \mathbb{R}^k$ by

$$\mathbf{y} = \mathrm{FC}(\mathbf{x}) = \mathbf{W}\mathbf{x} + \boldsymbol{b}, \quad (3.1)$$

where the weight matrix of parameters $\mathbf{W} \in \mathbb{R}^{k \times d}$ and the bias vector of parameters $\boldsymbol{b} \in \mathbb{R}^k$. The FC layer in (3.1) is fundamental to many modern and traditional neural network architectures. Note that the degree of freedom for the weight parameters $\mathbf{W}$ in (3.1) is $kd$. Since $\mathbf{W}$ dominates parameterization, the parameter size of the FC layer in (3.1) is $\mathcal{O}(kd)$.

## 3.2    PARAMETERIZED HYPERCOMPLEX MULTIPLICATION (PHM) LAYERS

We propose the parameterized hypercomplex multiplication (PHM) layer that transforms an input $\mathbf{x}$ into an output $\mathbf{y}$ by

$$\mathbf{y} = \mathrm{PHM}(\mathbf{x}) = \mathbf{H}\mathbf{x} + \boldsymbol{b}, \quad (3.2)$$

where the same notation from (3.1) is used but the replaced parameter $\mathbf{H} \in \mathbb{R}^{k \times d}$ is constructed by a sum of Kronecker products. For context, the Kronecker product is a generalization of the vector outer product to higher dimensions in real space. For any matrix $\mathbf{X} \in \mathbb{R}^{m \times n}$ and $\mathbf{Y} \in \mathbb{R}^{p \times q}$, the Kronecker product $\mathbf{X} \otimes \mathbf{Y}$ is a block matrix:

$$\mathbf{X} \otimes \mathbf{Y} = \begin{bmatrix} x_{11}\mathbf{Y} & \dots & x_{1n}\mathbf{Y} \\ \vdots & \ddots & \vdots \\ x_{m1}\mathbf{Y} & \dots & x_{mn}\mathbf{Y} \end{bmatrix} \in \mathbb{R}^{mp \times nq},$$

where $x_{ij}$ is the element of $\mathbf{X}$ at its $i^{\text{th}}$ row and $j^{\text{th}}$ column. Note that the symbol $\otimes$ between two matrices is the Kronecker product while the same symbol between two quaternions means the Hamilton product.

Now let us revisit (3.2) to explain $\mathbf{H}$. Suppose that both $k$ and $d$ are divisible by a user-defined hyperparameter $n \in \mathbb{Z}_{>0}$. For $i = 1, \dots, n$, denote by each parameter matrix $\mathbf{A}_i \in \mathbb{R}^{n \times n}$ and $\mathbf{S}_i \in \mathbb{R}^{\frac{k}{n} \times \frac{d}{n}}$. The parameter $\mathbf{H}$ in (3.2) is a sum of $n$ Kronecker products:

$$\mathbf{H} = \sum_{i=1}^{n} \mathbf{A}_i \otimes \mathbf{S}_i. \quad (3.3)$$

Parameters for $\mathbf{H}$:          Size of $\mathbf{H}$:     Parameter size of $\mathbf{H}$:

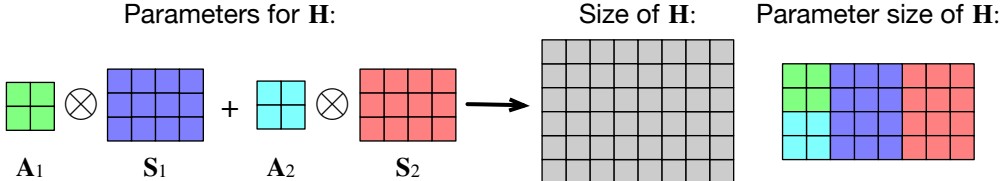

Figure 1: Illustration of the PHM layer. It uses a sum of Kronecker products of matrices $\mathbf{A}_i$ and $\mathbf{S}_i$ ($i = 1, 2$) to construct $\mathbf{H}$ in (3.2) (here $n = 2, k = 6, d = 8$). Best viewed in color.

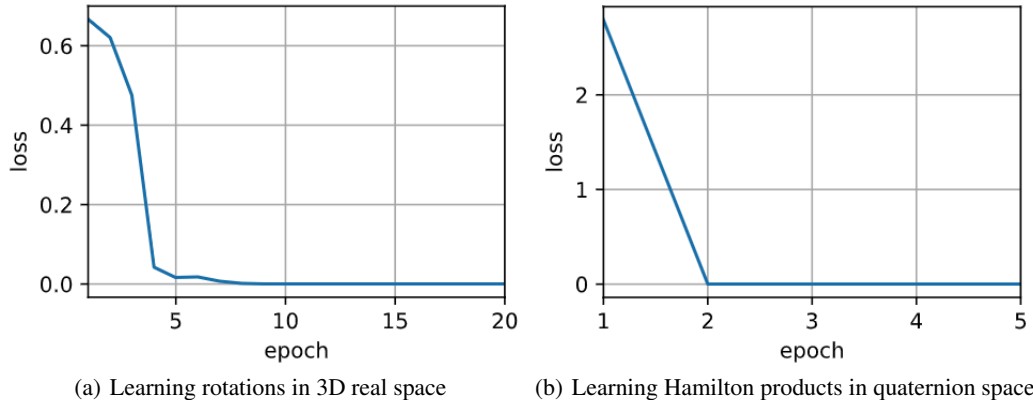

(a) Learning rotations in 3D real space          (b) Learning Hamilton products in quaternion space

Figure 2: PHM layers can learn to perform rotations in 3D real space and Hamilton products in quaternion space on artificial datasets.

As illustrated in Figure 1, it is the parameter matrices $\mathbf{A}_i$ and $\mathbf{S}_i$ ($i = 1, \ldots, n$) that determine the degree of freedom for $\mathbf{H}$, which is $kd/n + n^3$. Since $\mathbf{H}$ dominates parameterization, the parameter size of the PHM in (3.2) is $\mathcal{O}(kd/n)$, where $kd \gtrsim n^4$ is assumed: this condition is mild for real-world problems, such as in our experiments (e.g., $d = 512$, $k = 2048$, $n = 2, 4, 8, 16$). Thus, for the same input and output sizes, the parameter size of a PHM layer is approximately $1/n$ of that of an FC layer under mild assumptions.

The benefit of parameterization reduction of PHM layers is due to reusing elements of both parameter matrices $\mathbf{A}_i$ and $\mathbf{S}_i$ in the Kronecker product. As an alternative perspective, we can equivalently reconstruct $\mathbf{H}$ in (3.3) by reusing parameter matrices in real-valued matrix multiplications, followed by more operations. Due to limited space, this more complicated perspective is offered in Appendix A. Though simply setting $\mathbf{H} = \mathbf{A}_1 \otimes \mathbf{S}_1$ can further save parameters, it does not generalize hypercomplex multiplications hence is out of scope.

To show that PHM layers can learn to perform pre-defined multiplication-related operations in practice, we perform experiments to learn rotations in 3D real space using the PHM layer. Using a rotation matrix $\mathbf{W} \in \mathbb{R}^{3 \times 3}$ we create an artificial dataset $\{(\mathbf{x}_i \in \mathbb{R}^3, \mathbf{y}_i \in \mathbb{R}^3)\}$, where $\mathbf{y}_i$ is generated via the 3D rotation of the input: $\mathbf{y}_i = \mathbf{W}\mathbf{x}_i$. Figure 2(a) shows that the loss converges to zero: the PHM layer can learn a single rotation of an object in 3D real space.

In the following, we show how the proposed PHM layer subsumes and generalizes both hypercomplex multiplications and real-valued matrix multiplications.

### 3.3 SUBSUMING HYPERCOMPLEX MULTIPLICATIONS

First, we explore how the PHM layer connects to the hypercomplex multiplication. For the sake of illustration, let us take the Hamilton product of two quaternions $Q$ and $P$ in (2.2) as an example,

which can be rewritten as

$$\begin{bmatrix} Q_r & -Q_x & -Q_y & -Q_z \\ Q_x & Q_r & -Q_z & Q_y \\ Q_y & Q_z & Q_r & -Q_x \\ Q_z & -Q_y & Q_x & Q_r \end{bmatrix} \begin{bmatrix} P_r \\ P_x \\ P_y \\ P_z \end{bmatrix}, \tag{3.4}$$

where the 4 output elements are the real values for the quaternion unit basis $[1, \mathbf{i}, \mathbf{j}, \mathbf{k}]^\top$. Note that for models leveraging Hamilton products of quaternions (Parcollet et al., 2018b; 2019; Tay et al., 2019), the components $Q_r, Q_x, Q_y, Q_z$ of (3.4) are learnable parameters while the components $P_r, P_x, P_y, P_z$ are the layer inputs. In practice, such a layer usually has more than 4 inputs ($d > 4$). To apply the Hamilton product, all the inputs are evenly split into 4 segments ($P_r, P_x, P_y, P_z$) of the right input vector of (3.4). Then each component in the left matrix of (3.4) can be a block matrix (i) where all the elements take the same value; (ii) whose shape is aligned with the input length $d$ and the output length $k$ of the layer. It is noteworthy that the left $4 \times 4$ matrix of (3.4) can be rewritten as a sum of 4 Kronecker products:

$$\underbrace{\begin{bmatrix} 1 & 0 & 0 & 0 \\ 0 & 1 & 0 & 0 \\ 0 & 0 & 1 & 0 \\ 0 & 0 & 0 & 1 \end{bmatrix}}_{\mathbf{A}_1} \otimes \underbrace{[Q_r]}_{\mathbf{S}_1} + \underbrace{\begin{bmatrix} 0 & -1 & 0 & 0 \\ 1 & 0 & 0 & 0 \\ 0 & 0 & 0 & -1 \\ 0 & 0 & 1 & 0 \end{bmatrix}}_{\mathbf{A}_2} \otimes \underbrace{[Q_x]}_{\mathbf{S}_2} + \underbrace{\begin{bmatrix} 0 & 0 & -1 & 0 \\ 0 & 0 & 0 & 1 \\ 1 & 0 & 0 & 0 \\ 0 & -1 & 0 & 0 \end{bmatrix}}_{\mathbf{A}_3} \otimes \underbrace{[Q_y]}_{\mathbf{S}_3} + \underbrace{\begin{bmatrix} 0 & 0 & 0 & -1 \\ 0 & 0 & -1 & 0 \\ 0 & 1 & 0 & 0 \\ 1 & 0 & 0 & 0 \end{bmatrix}}_{\mathbf{A}_4} \otimes \underbrace{[Q_z]}_{\mathbf{S}_4}.$$

$$(3.5)$$

According to (3.5), when $n = 4$, the PHM layer can be learned to express the Hamilton product of quaternions. Specifically, matrices $\mathbf{A}_1, \ldots, \mathbf{A}_4$ in (3.3) parameterize the four matrices composed of $-1, 0, 1$ in (3.5) that reflect interactions between real and imaginary components of quaternions, which are the rule of Hamilton products. The single-element "matrices" $\mathbf{S}_1, \ldots, \mathbf{S}_4$ in (3.3) are equal to the learnable components $Q_r, Q_x, Q_y, Q_z$ in (3.4). Figure 2(b) shows that PHM layers can learn the rule of Hamilton products on artificial data. Likewise, hypercomplex multiplications of octonions or sedenions can also be learned by the PHM layer when $n$ is set to 8 or 16.

### 3.4 SUBSUMING REAL-VALUED MATRIX MULTIPLICATIONS

Next, we show how the PHM layer subsumes the matrix multiplication in real space. In other words, the PHM layer is a generalization of the FC layer via the hyperparameter $n$. To explain, referring to (3.2), when $n = 1$, $\mathbf{H} = \mathbf{A}_1 \otimes \mathbf{S}_1 = a\mathbf{S}_1$, where the scalar $a$ is the single element of the $1 \times 1$ matrix $\mathbf{A}_1$ and $\mathbf{S}_1 \in \mathbb{R}^{k \times d}$. Since learning $a$ and $\mathbf{S}_1$ separately is equivalent to learning their multiplication jointly, scalar $a$ can be dropped, which is learning the single weight matrix in an FC layer. Therefore, a PHM layer is degenerated to an FC layer when $n = 1$.

### 3.5 GENERALIZING HYPERCOMPLEX MULTIPLICATIONS

Though parameter reusing by component-wise partitioning in quaternion space has demonstrated success (Parcollet et al., 2018b; Zhu et al., 2018; Parcollet et al., 2019; Tay et al., 2019), one key problem is that hypercomplex space only exists at very few predefined dimensionalities, such as 4D (quaternions), 8D (octonions), and 16D (sedenions). Within the context of hypercomplex space, specialized multiplication rules, such as the Hamilton product, have to be devised and encoded in the network as a fixed inductive bias. As described in Section 1, the very few choices over existing hypercomplex space restricts the flexibility of networks that leverage hypercomplex multiplication.

In sharp contrast to relying on predefined mathematical rules over limited dimensionality choices, the PHM layer treats the dimensionality $n$ (number of Kronecker products) as a tunable hyperparameter and learns such specialized multiplication rules from data, as manifested in the parameterized matrices $\mathbf{A}_i$ ($i = 1, \ldots, n$) in (3.3). On one hand, the PHM layer can express hypercomplex multiplications when $\mathbf{A}_i$ are set to reflect those predefined multiplication rules in hypercomplex space. On the other hand, the PHM layer can be seen as a trainable and parameterized form of $n$D hypercomplex multiplications, where $n$ can be values other than 4, 8, or 16. Thus, the PHM layer generalizes multiplications in hypercomplex space. Since $n$ can be 1, the PHM layer also offers a neat way to bridging multiplication between both real space and hypercomplex space.

## 4 NEURAL MODELS WITH PHM LAYERS

To demonstrate the applicability of the PHM layers, we develop the PHM-LSTM and PHM-transformer by equipping two popular neural network models, LSTMs and transformers, with PHM layers.

### 4.1 PHM-LSTM

Recurrent neural networks such as LSTMs (Hochreiter & Schmidhuber, 1997) are gated recurrent networks where the gating functions are parameterized by linear transformations. We introduce the PHM-LSTM, which replaces such linear transformations in LSTMs with PHM layers:

$$\mathbf{y}_t = \text{PHM}\,(\mathbf{x}_t) + \text{PHM}\,(\mathbf{h}_{t-1}) + \boldsymbol{b}$$
$$\mathbf{f}_t, \mathbf{i}_t, \mathbf{o}_t, \mathbf{x}'_t = \phi(\mathbf{y}_t)$$
$$\mathbf{c}_t = \sigma_s(\mathbf{f}_t)\,\mathbf{c}_{t-1} + \sigma_s(\mathbf{i}_t)\,\sigma_t(\mathbf{x}'_t)$$
$$\mathbf{h}_t = \mathbf{o}_t \odot \mathbf{c}_t,$$

where $\sigma_s$ is the sigmoid activation function, $\sigma_t$ is the tanh activation function, $\phi : \mathbb{R}^{1 \times d} \to \mathbb{R}^{4 \times \frac{d}{4}}$ is a four-way split on the last dimension, and $\mathbf{c}_t, \mathbf{h}_t$ are the cell state and the hidden state of the PHM-LSTM unit at any time step $t$.

### 4.2 PHM-TRANSFORMER

The transformer is a stacked neural network architecture that aggressively exploits linear transformations (Vaswani et al., 2017). Each self-attention layer comprises of $\mathbf{Q}$ (query), $\mathbf{K}$ (key), $\mathbf{V}$ (value) linear transformations, along with multiple heads. Each transformer block also has a position-wise feed-forward network composed of two FC layers. Since a large majority of the transformer parameters stem from linear transformations or FC layers, we introduce the PHM-transformer to replace all the linear transformations or FC layers with PHM layers. The single-head self-attention module is rewritten as:

$$\mathbf{Q}, \mathbf{K}, \mathbf{V} = \Phi(\text{PHM}(\mathbf{X}))$$
$$\mathbf{A} = \text{softmax}\big(\frac{\mathbf{Q}\mathbf{K}^\top}{\sqrt{d_k}}\big)\mathbf{V},$$

where $d_k$ is the key dimension, $\Phi : \mathbb{R}^{1 \times d} \to \mathbb{R}^{3 \times \frac{d}{3}}$ is a three-way split on the last dimension, $\mathbf{X}$ is the input sequence, and $\mathbf{A}$ is the self-attentive representation. For multi-head attention, using PHM layers also enables weight sharing not only among the linear transformations of $\mathbf{Q}, \mathbf{K}, \mathbf{V}$ but also among the linear transformation of multiple heads:

$$\mathbf{X} = \text{PHM}([\mathbf{H}_1; \ldots; \mathbf{H}_{N_h}]),$$

where $N_h$ is the number of heads and (;) is the column-wise concatenation. Finally, the position-wise feed-forward network is now defined as

$$\mathbf{Y} = \text{PHM}(\text{ReLU}(\text{PHM}(\mathbf{X}))),$$

which transforms $\mathbf{X}$ with two PHM layers.

## 5 EXPERIMENTS

For context, in the field of representation learning using hypercomplex multiplications, quaternion convolutional neural networks (Zhu et al., 2018), quaternion recurrent neural networks (Parcollet et al., 2018a), and quaternion transformers (Tay et al., 2019) have all compared themselves with only real-valued counterparts. Therefore, to be consistent with the rest of the literature, we evaluate PHM-LSTMs and PHM-transformers that are equipped with PHM layers, and compare them with quaternion LSTMs, quaternion transformers, real-valued LSTMs, or real-valued transformers. Both quaternion LSTMs and quaternion transformers replace linear transformations with Hamilton products of quaternions.

Table 1: Experimental results of natural language inference (accuracy) on five different datasets. The PHM-LSTM reduces the parameters of the standard LSTM model and improves or partially matches performance on four out of five datasets.

| Model | #Params | MNLI | QNLI | SNLI | DNLI | SciTail |
|-------|---------|------|------|------|------|---------|
| LSTM | 721K | **71.82** / 71.89 | 84.44 | 84.18 | 85.16 | 74.36 |
| Quaternion LSTM | 180K (-75.0%) | 71.57 / **72.19** | **84.73** | 84.21 | 86.45 | 75.58 |
| PHM-LSTM ($n = 2$) | 361K (-49.9%) | **71.82** / 72.08 | 84.39 | 84.38 | 85.77 | 77.47 |
| PHM-LSTM ($n = 5$) | 146K (-79.7%) | 71.80 / 71.77 | 83.87 | **84.58** | **86.47** | 74.64 |
| PHM-LSTM ($n = 10$) | 81K (-88.7%) | 71.59 / 71.59 | 84.25 | 84.40 | 86.21 | **77.84** |

To demonstrate the architectural flexibility and effectiveness, we evaluate different settings of PHM-LSTMs and PHM-transformers to show that allowing for flexible choices of the hyperparameter $n$ in the PHM layer may lead to more effective performance. Details of the setup for the experiments are provided in Appendix B.

## 5.1 NATURAL LANGUAGE INFERENCE

The task of natural language inference is to determine the logical relationship between two text sequences (MacCartney, 2009). It is a fundamental task pertaining to language understanding. To this end, they serve as a suitable benchmark for evaluating recurrent models.

We run experiments on five datasets: (i) MultiNLI (Williams et al., 2017), (ii) QNLI (Quora) (Wang et al., 2017), (iii) SNLI (Bowman et al., 2015), (iv) Dialogue NLI (Welleck et al., 2018), and (v) SciTail (Science Entailment) (Khot et al., 2018). Table 1 reports the results on all these datasets. All in all, such results show that the PHM layer can not only reduce the parameters but also improve performance with flexible choices of $n$ (four out of five datasets show reasonable improvement or partially match). The only exception is on the QNLI dataset, where the performance drop is marginal ($< 1\%$). This is still decent considering the parameter saving: the parameterization cost of the PHM-LSTM is in the order of $\mathcal{O}(1/n)$ of that of the standard LSTM, where settings of $n = 5$ and $n = 10$ do not take values of power of 2. As detailed in Appendix B, since we use the 300D GloVe (Pennington et al., 2014) embeddings to represent input tokens, we choose multiples of 5 instead of 4 for ease of divisibility. It is also noteworthy that on the SNLI, Dialogue NLI, and SciTail datasets, all the PHM-LSTM variants outperform the standard LSTM model. We think that the element reusing properties of the Kronecker product operation, in addition to learning to share such reused components amongst recurrent gating functions, may contribute to both effective and efficient representations.

## 5.2 MACHINE TRANSLATION

Machine translation is concerned with translating between source-target language pairs. To this end, sequence transduction models are central to this problem domain. In this experiment, the key goal is to compare PHM-transformers against the standard and quaternion transformer models.

We run experiments on seven datasets: (i) IWSLT'15 English-Vietnamese (En-Vi), (ii) IWSLT'17 English-Indonesian (En-Id), (iii) IWSLT'14 German-English (De-En), (iv) IWSLT'14 Romanian-English (Ro-En), (v) WMT'18 English-Estonian (En-Et), (vi) Setimes English-Macedonian (En-Mk), and (vii) WMT'18 English-Romanian (En-Ro).

Table 2 reports our results of the machine translation tasks. Overall, these empirical results with different settings demonstrate architectural flexibility and effectiveness of the hypercomplex multiplication parameterization. First and foremost, across six out of seven benchmarks, the PHM-transformer at $n = 4$ makes reasonable gains over the quaternion transformer, signifying that parameterization of hypercomplex multiplications by learning from data can be more effective than predefining Hamilton product rules mathematically. Second, though increasing $n$ leads to more parameter savings, we observe that increasing $n$ all the way to 16 does not cause significant degradation in performance on datasets such as En-Vi. Third, for most datasets, even with significant parameter savings, we find that the decrease in the BLEU score is mostly manageable ($\approx 1$–3 BLEU

Table 2: Experimental results of machine translation (BLEU) on seven different datasets. Symbol † represents re-scaling the parameters with a factor of 2 by doubling the hidden size. The PHM-transformer does not lose much performance despite enjoying parameter savings. Re-scaling can lead to improvement in performance.

| Model | #Params | En-Vi | En-Id | De-En | Ro-En | En-Et | En-Mk | En-Ro |
|---|---|---|---|---|---|---|---|---|
| Transformer (Tm) | 44M | 28.43 | 47.40 | **36.68** | **34.60** | 14.17 | 13.96 | **22.79** |
| Quaternion Tm | 11M (-75.0%) | 28.00 | 42.22 | 32.83 | 30.53 | 13.10 | 13.67 | 18.50 |
| PHM-Tm $n = 2$ | 22M (-50.0%) | 29.25 | 46.32 | 35.52 | 33.40 | **14.98** | 13.60 | 21.73 |
| PHM-Tm $n = 4$ | 11M (-75.0%) | 29.13 | 44.13 | 35.53 | 32.74 | 14.11 | 13.01 | 21.19 |
| PHM-Tm $n = 8$ | 5.5M (-87.5%) | 29.34 | 40.81 | 34.16 | 31.88 | 13.08 | 12.95 | 21.66 |
| PHM-Tm $n = 16$ | 2.9M (-93.4%) | 29.04 | 33.48 | 33.89 | 31.53 | 12.15 | 11.97 | 19.63 |
| PHM-Tm† $n = 2$ | 44M | **29.54** | **49.05** | 34.32 | 33.88 | 14.05 | **14.41** | 22.18 |
| PHM-Tm† $n = 4$ | 22M (-50.0%) | 29.17 | 46.24 | 34.86 | 33.80 | 14.43 | 13.78 | 21.91 |
| PHM-Tm† $n = 8$ | 11M (-75.0%) | 29.47 | 43.49 | 34.71 | 32.59 | 13.75 | 13.78 | 21.43 |

Table 3: Training time (seconds per 100 steps) and inference time (seconds to decode test sets) with beam size of $4$ and length penalty of $0.6$ on the IWSLT'14 German-English dataset.

| Model | Transformer (Tm) | Quaternion Tm | PHM-Tm ($n = 4$) | PHM-Tm ($n = 8$) |
|---|---|---|---|---|
| Training time | **7.61** | 8.11 | 7.92 | 7.70 |
| Inference time | 336 | 293 | 299 | **282** |

points). However, we also note a rare occurrence where $n = 16$ results in a significant decrease in the BLEU score, such as on the En-Id dataset. Fourth, on several datasets, the PHM-transformer model improves the performance of the standard transformer model. For example, on datasets such as En-Vi and En-Et, the PHM-transformer model enjoys a performance boost of about $0.8$ BLEU point with $n = 2$. Finally, by re-scaling with a factor of 2 (doubling the hidden size), we are able to improve the performance on three datasets: En-Vi, En-Id, and En-Mk.

Table 3 reports the training and inference time for transformer variants. We observe that the PHM-transformer with $n = 8$ has the fastest inference speed amongst all the variants, primarily due to a significant reduction of parameters. All in all, the training speed is also approximately comparable. This ascertains that the PHM layer does not increase much computational cost in practice.

### 5.3 TEXT STYLE TRANSFER

We continue to experiment with sequence transduction for text style transfer. The goal of this task is to convert text of a certain style to another style. We use the Modern→Shakespeare corpus[1] in the experiments. Table 4 reports the results on this text style transfer task. We observe that the best performance is achieved with the PHM-transformer ($n = 4$). Notably, all except the $n = 16$ variant increases or matches the performance of the standard transformer model. This ascertains architectural flexibility and effectiveness of the proposed PHM layer. This not only enables parameter savings but also improves the performance of the transformer.

### 5.4 SUBJECT VERB AGREEMENT

We conduct additional experiments on the subject-verb agreement task (Linzen et al., 2016). The task predicts if the sentence, e.g., *'The keys to the cabinet _____ .'* is followed by a plural or a singular. The used dataset can be found online (Linzen et al., 2016). Table 5 reports the results on the subject-verb agreement task. Results are promising, demonstrating that all variants with PHM layers outperform the standard and quaternion transformer models. The best performance peaks at $n = 8$, despite a parameter saving to up to $1/8$.

---

[1] `https://github.com/tlatkowski/st`

Table 4: Experimental results of text style transfer. The PHM-transformer may reduce the parameters of the standard transformer model and improve performance.

| Model | #Params | BLEU |
|---|---|---|
| Transformer (Tm) | 44M | 11.65 |
| PHM-Tm ($n = 2$) | 22M (-50.0%) | 12.20 |
| PHM-Tm ($n = 4$) | 11M (-75.0%) | **12.42** |
| PHM-Tm ($n = 8$) | 5.5M (-87.5%) | 11.66 |
| PHM-Tm ($n = 16$) | 2.9M (-93.4%) | 10.76 |

Table 5: Experimental results of subject verb agreement. The PHM-transformer may reduce the parameters of the standard transformer model and improve performance.

| Model | #Params | Acc |
|---|---|---|
| Transformer (Tm) | 400K | 94.80 |
| Quaternion Tm | 100K | 94.70 |
| PHM-Tm ($n = 2$) | 200K (-50.0%) | 95.14 |
| PHM-Tm ($n = 4$) | 101K (-74.8%) | 95.05 |
| PHM-Tm ($n = 8$) | 56K (-86.0%) | **95.62** |

## 6 RELATED WORK

While neural networks have been a well-established line of research, progress on hypercomplex representations for deep learning is still in its infancy and most works on this topic are new (Gaudet & Maida, 2017; Parcollet et al., 2018a;b; Zhu et al., 2018; Tay et al., 2019). The hypercomplex Hamilton product provides a greater extent of expressiveness, similar to the complex multiplication, albeit with a 4-fold increase in interactions between real and imaginary components. In the case of quaternion representations, due to parameter savings in the Hamilton product, models also enjoy a 75% reduction in the parameter size (Parcollet et al., 2018a; Tay et al., 2019). A striking caveat is that all quaternions are fundamentally limited to 4D hypercomplex space, which restricts architectural flexibility. The other options would be to scale to octonion (8D) or sedenion (16D) space, given the predefined multiplication rules in such space. To the best of our knowledge, there is no work that attempts to generalize arbitrary $n$D hypercomplex multiplications to allow for architectural flexibility, where $n$ can be specified or tuned by users.

Our work can also be interpreted as a form of soft parameter sharing, albeit learned from data. Quaternion networks (Zhu et al., 2018; Parcollet et al., 2018b; 2019) are known to possess weight sharing properties via the Hamilton product operation and have demonstrated reasonable success despite having fewer parameters. To the best of our knowledge, there has been no work that attempts to parameterize the hypercomplex Hamilton product for neural networks, i.e., enabling end-to-end learning of real and imaginary component interactions from data.

## 7 CONCLUSION

We proposed parameterized hypercomplex multiplication (PHM) layers that learn and generalize hypercomplex multiplications. In practice, the PHM layer has $1/n$ learnable parameters compared with the fully-connected layer counterpart, where $n$ can be flexibly specified by users. PHM layers are applicable to dominant models such as LSTMs and transformers. We evaluated these models equipped by PHM layers on comprehensive tasks to show architectural flexibility and effectiveness of the hypercomplex multiplication parameterization.

**Acknowledgements.** We thank the anonymous reviewers for the insightful comments on this paper. This work was partially supported by the Ministry of Education (MoE) of Singapore under the Academic Research Fund (AcRF) Tier 1 Grant RG135/18.

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

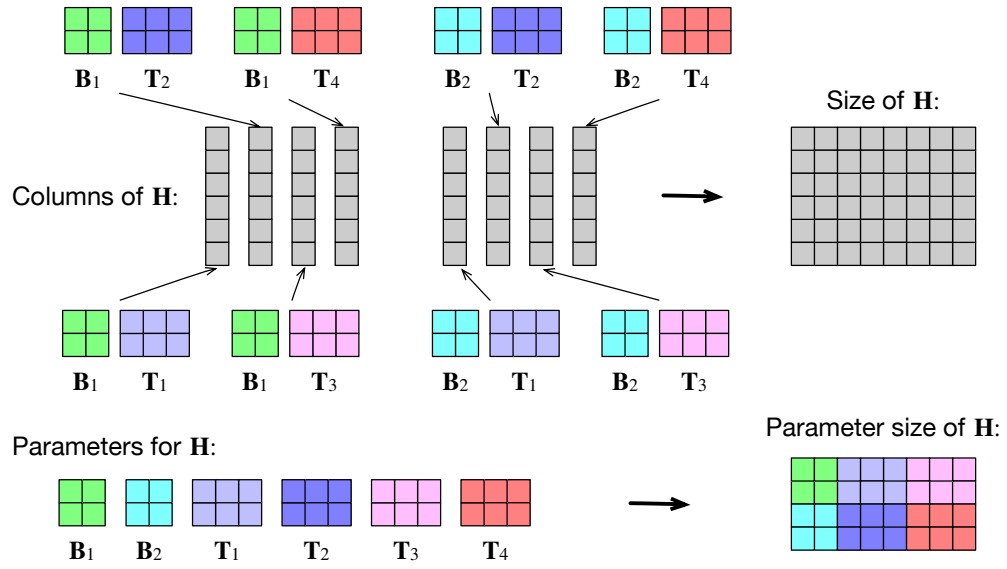

Figure 3: Illustration of reconstructing $\mathbf{H}$ in (3.2) by reusing parameter matrices $\mathbf{B}_i$ ($i = 1, 2$) and $\mathbf{T}_j$ ($j = 1, \ldots, 4$) in real-valued matrix multiplications, followed by more operations (here $n = 2, k = 6, d = 8$). Best viewed in color.

## A    RECONSTRUCTING THE PARAMETER MATRIX

In the paper, the parameter matrix $\mathbf{H}$ in (3.2) is constructed by a sum of $n$ Kronecker products. In the following, we will provide an alternative perspective and show how to equivalently reconstruct $\mathbf{H}$ by reusing parameter matrices in real-valued matrix multiplications, followed by more operations.

### A.1    METHOD

The key idea is to operate on partitioned weight blocks and learn a dynamic diffusion of weights. There are two key parameter blocks $\mathbf{B}$ and $\mathbf{T}$ that are central to our approach. Intuitively, $\mathbf{B} \in \mathbb{R}^{n \times n \times n}$ controls the weight diffusion process and learns the soft interactions between $\mathbf{T}$ partitions. Here, $n$ is a user defined hyperparameter.

Suppose that both $d$ and $k$ are divisible by $n \in \mathbb{Z}_{>0}$. For $i = 1, \ldots, n$ and $j = 1, \ldots, \frac{d}{n}$, denote by each partitioned parameter block $\mathbf{T}_j \in \mathbb{R}^{n \times \frac{k}{n}}$, and $\mathbf{B}_i \in \mathbb{R}^{n \times n}$ is the weight diffusion matrix assigned to each partitioned parameter block via real-valued matrix multiplication $\mathbf{B}_i \mathbf{T}_j$. The parameter $\mathbf{H}$ in (3.2) is now constructed by column-wise concatenation (;):

$$\mathbf{H} = [s(\mathbf{B}_1); s(\mathbf{B}_2); \ldots; s(\mathbf{B}_n)], \tag{A.1}$$

where each segment $s(\mathbf{B}_i)$ is also formed by column-wise concatenation:

$$s(\mathbf{B}_i) = [\psi(\mathbf{B}_i \mathbf{T}_1); \psi(\mathbf{B}_i \mathbf{T}_2); \ldots; \psi(\mathbf{B}_i \mathbf{T}_{\frac{d}{n}})]. \tag{A.2}$$

In (A.2), function $\psi : \mathbb{R}^{p \times q} \to \mathbb{R}^{pq}$, where $\psi(\mathbf{X})$ flattens the matrix $\mathbf{X} \in \mathbb{R}^{p \times q}$ by concatenating each row of $\mathbf{X}$ then transposes the concatenated row vector into a column vector of dimension $pq$. It is easy to see that, $\psi(\mathbf{B}_i \mathbf{T}_j) \in \mathbb{R}^k$, $s(\mathbf{B}_i) \in \mathbb{R}^{k \times \frac{d}{n}}$, thus $\mathbf{H} \in \mathbb{R}^{k \times d}$.

It is the partitioned parameter blocks $\mathbf{B}_i$ ($i = 1, \ldots, n$) and $\mathbf{T}_j$ ($j = 1, \ldots, \frac{d}{n}$) that determine the degree of freedom for $\mathbf{H}$, which is $kd/n + n^3$. As illustrated in Figure 3, the reuse of parameter matrices $\mathbf{B}_1, \ldots, \mathbf{B}_n$ and $\mathbf{T}_1, \ldots, \mathbf{T}_{\frac{d}{n}}$ in real-valued matrix multiplications in (A.2) may reduce the degree of freedom for $\mathbf{H}$.

## A.2 Subsuming Hypercomplex Multiplications

Similarly, we show how the PHM layer with the reconstructed $\mathbf{H}$ in (A.1) also subsumes the hypercomplex multiplication. Taking the Hamilton product of two quaternions $Q$ and $P$ as an example, it can be rewritten as

$$
\left( \underbrace{\begin{bmatrix} 1 & 0 & 0 & 0 \\ 0 & 1 & 0 & 0 \\ 0 & 0 & 1 & 0 \\ 0 & 0 & 0 & 1 \end{bmatrix}}_{\mathbf{B}_1} \underbrace{\begin{bmatrix} Q_r \\ Q_x \\ Q_y \\ Q_z \end{bmatrix}}_{\mathbf{T}_1} ; \underbrace{\begin{bmatrix} 0 & -1 & 0 & 0 \\ 1 & 0 & 0 & 0 \\ 0 & 0 & 0 & 1 \\ 0 & 0 & -1 & 0 \end{bmatrix}}_{\mathbf{B}_2} \underbrace{\begin{bmatrix} Q_r \\ Q_x \\ Q_y \\ Q_z \end{bmatrix}}_{\mathbf{T}_1} ; \underbrace{\begin{bmatrix} 0 & 0 & -1 & 0 \\ 0 & 0 & 0 & -1 \\ 1 & 0 & 0 & 0 \\ 0 & 1 & 0 & 0 \end{bmatrix}}_{\mathbf{B}_3} \underbrace{\begin{bmatrix} Q_r \\ Q_x \\ Q_y \\ Q_z \end{bmatrix}}_{\mathbf{T}_1} ; \underbrace{\begin{bmatrix} 0 & 0 & 0 & -1 \\ 0 & 0 & 1 & 0 \\ 0 & -1 & 0 & 0 \\ 1 & 0 & 0 & 0 \end{bmatrix}}_{\mathbf{B}_4} \underbrace{\begin{bmatrix} Q_r \\ Q_x \\ Q_y \\ Q_z \end{bmatrix}}_{\mathbf{T}_1} \right) \begin{bmatrix} P_r \\ P_x \\ P_y \\ P_z \end{bmatrix},
$$
(A.3)

where the 4 output elements are the real values for the quaternion unit basis $[1, \mathbf{i}, \mathbf{j}, \mathbf{k}]^\top$. According to (A.3), when $n = 4$, the PHM layer with the reconstructed parameter matrix can also be learned to exactly express the Hamilton product of quaternions. Likewise, hypercomplex multiplications of octonions or sedenions can also be learned by the PHM layer when $n$ is set to 8 or 16.

## A.3 Subsuming Real-Valued Matrix Multiplications

Now we show how the PHM layer with the reconstructed $\mathbf{H}$ in (A.1) also subsumes the matrix multiplication in real space. Referring to (3.2), when $n = 1$, $\mathbf{H} = b\mathbf{W}$, where the scalar $b$ is the single element of the $1 \times 1$ matrix $\mathbf{B}_1$ and elements of $\mathbf{W} \in \mathbb{R}^{k \times d}$ come from the concatenation of $\mathbf{T}_1, \ldots, \mathbf{T}_d \in \mathbb{R}^{1 \times k}$. Since learning $b$ and $\mathbf{W}$ separately is equivalent to learning their multiplication jointly, the scalar $b$ can be dropped, which is learning the single weight matrix in an FC layer. Therefore, a PHM layer is degenerated to an FC layer when $n = 1$.

# B    Setup for Experiments

We describe the setup for the experiments as follows.

## B.1 Natural Language Inference

We implement 300D unidirectional encoders with shared parameters for both premises and hypotheses. We take the concatenation of max and mean pooled representations as the input to a two-layer 300D multilayer perceptron for prediction. Our model is trained with the the Adam optimizer with a learning rate of $0.0004$ and a batch size of $256$. Word embeddings are initialized with GloVe (Pennington et al., 2014) and are fixed. No cross sentence attention (Parikh et al., 2016) is used, mainly to observe the effectiveness of standalone encoders. For PHM-LSTM, we use $n = \{2, 5, 10\}$. Note that in this task, since word embeddings are 300D, we select multiples of $5$ instead of $4$ for ease of divisibility.

## B.2 Machine Translation

For the IWSLT'15 English-Vietnamese (En-Vi), IWSLT'17 English-Indonesian (En-Id), IWSLT'14 German-English (De-En), and IWSLT'14 Romanian-English (Ro-En) datasets, we run with 50K steps; while for WMT'18 English-Estonian (En-Et), Setimes English-Macedonian (En-Mk), and WMT'18 English-Romanian (En-Ro) datasets, models are trained for 100K steps. For the En-Vi, En-Id, En-Et, En-Mk, and En-Ro datasets, we specify that transformers have 4 layers, 8 heads, and a hidden size 512. For the De-En and Ro-En datasets, we specify that transformers have 2 layers, 4 heads, and a hidden size 256. We use beam size of $5$ and $\alpha = 0.6$ (length penalty) for decoding. For all PHM models, we benchmark several settings for the hyperparameter $n = \{2, 4, 8, 16\}$.

### B.3 Text Style Transfer

For the used Modern→Shakespeare corpus[2] in the experiments, the key goal here is to convert modern writing into Shakespeare writing. This dataset comprises of $18,395$ parallel sentences for training, $1,218$ parallel sentences for evaluation (development set), and $1,462$ parallel sentences for testing. We still specify that transformers have 4 layers, 8 heads, and a hidden size 512. Similar to machine translation, we experiment with $n = \{2, 4, 8, 16\}$. We train all the models for 10K steps.

### B.4 Subject Verb Agreement

In contrast to the previous experimental settings, we use a smaller transformer architecture with 10K training steps. Specifically, transformers here have 2 layers, 4 heads, and a hidden size 128. Since the hidden size is smaller than those in the previous experimental settings, we experiment with $n = \{2, 4, 8\}$.

---

[2]`https://github.com/tlatkowski/st`

