# OpenReview forum: "Beyond Fully-Connected Layers with Quaternions: Parameterization of Hypercomplex Multiplications with $1/n$ Parameters"
_ICLR.cc/2021/Conference — ICLR 2021 Spotlight_

### Official Review · AnonReviewer3 · 2020-10-28
**Solid contribution toward making hypercomplex operations more flexible**

**Rating:** 8
**Confidence:** 3

**Review:**

The authors focus on the area of using hypercomplex multiplications (multiplications involving numbers with multiple imaginary components) in deep learning models. Past work in this area has been promising but has been limited to certain dimensions for which there are predefined multiplication operations. The novel contribution of this work is to parameterize the hypercomplex multiplication operations, enabling the model to discover new operations rather than relying on the small number of existing operations and the small number of dimensions for which such operations exist. The authors find that their approach can substantially reduce the number of parameters without reducing performance (and in some cases even improving performance).

Strengths:

1. The proposed method makes a promising approach from the literature more flexible, helping to pave the way for making this approach more broadly useful.

2. The authors illustrate this flexibility by showing how their approach can be effective for two different architectures (LSTMs and Transformers), making the general point that it can be applied to any architecture that uses feedforward components. They also apply it to multiple tasks, again illustrating the flexibility.

3. As mentioned above, the approach can substantially increase a model’s parameter count without affecting performance. Relatedly, it can also improve inference speed.

4. The paper is generally thorough and clear.

Weaknesses:

1. The specific contribution of this paper is the parameterization of the multiplication operation, but the evidence that this parameterization is helpful is mild, as there are only a few cases where the proposed model noticeably outperforms the Quaternion model. Thus, the evidence presented does not make a strong case for the necessity of this parameterization.

2. Much of the argument hinges on the reduced parameter count, but there was not any mention of exactly how many parameters each model had (at least, not that I saw - I did not check the appendix). I think the paper could be substantially strengthened by adding a “Parameter count” column to each table.

3. There is no clear intuition offered for why this approach might be expected to be effective. Offering such an intuition is certainly not necessary (since results alone are enough), but the paper would be more satisfying if there were such an intuition present.

Overall, I am rating this as a 7, because I find it to be a solid paper but worry that its contribution on top of the existing work that has studied hypercomplex operations may be too small and may not have enough evidence for its usefulness.

---

> ### Author Response · Authors · 2020-11-21
> **Response to Reviewer3**
>
>
> Thank you for the positive assessment and insightful comments.
>
> ### On “There Are Only a Few Cases where the Proposed Model Noticeably Outperforms the Quaternion Model”
> When comparing models at the same parameter saving level (75% saving), Table 2 has shown that our PHM-Transformer ($n=4$) outperforms Quaternion Transformer on 6 out of 7 machine translation tasks, and Table 5 has shown that our PHM-Transformer ($n=4$) outperforms Quaternion Transformer on the subject verb agreement task. Therefore, Section 5.2 highlights that results of Table 2 “signifying that parameterization of hypercomplex multiplications by learning from data can be more effective than predefining Hamilton product rules mathematically”.
>
>
>
> ### On Adding a Parameter Count Column to Each Table
> Thank you for the suggestions. We have revised the paper and included the #Params column in Table 1, 2, 4, 5.
>
>
>
> ### On Intuition for Why Our Method is Effective
> Thank you for considering providing this intuition as “not necessary”.
>
> Nonetheless, we still performed additional experiments on artificial datasets. First, we found that PHM can effectively learn predefined operations, such as a rotation in a 3D space. We have also added Section 4 in the revised supplementary materials to include such results. Second, our experiments on artificial datasets also show that PHM can learn to recover complex/Quaternion/Octonion/Sedenion multiplication rules (in the $\mathbf{A}$ matrices). Third, even for arbitrary dimensions where multiplication rules are undefined, when we manually specify multiplication rules to generate artificial datasets, PHM can still learn from the dataset to recover such rules. All these results suggest that PHM is both flexible and effective in learning different operations. Intuitively, the operations learned by PHM from data may be more effective than those predefined mathematical rules of hypercomplex spaces for the investigated tasks.

---

> > ### Comment · AnonReviewer3 · 2020-11-23
> > **Thank you for the response; score increased**
> >
> > Thank you for the response! After reviewing the data you have highlighted, along with the parameter counts you have added, I agree that the results are more impressive than I had initially concluded. Therefore, I have increased my score for this paper from 7 to 8.

---

> > > ### Author Response · Authors · 2020-11-24
> > > **Thanks for increasing the score!**
> > >
> > > Thank you and the other reviewers for the extremely helpful suggestions! We have also added the acknowledgements at the end of the revised paper.

---

### Official Review · AnonReviewer1 · 2020-10-30
**Interesting idea with practical benefits**

**Rating:** 8
**Confidence:** 4

**Review:**

The authors propose a novel way of parametrizing hypercomplex multiplications.
The proposed parametrization helps with: (a) Generalizing the multiplication to arbitrary dimensions, and (b) Reducing the number of parameters.
Building on this, the authors propose a parameterized hypercomplex multiplication (PHM) layer which essentially replaces the weight matrix of a linear layer with a matrix constructed via sum of Kronecker products.
They then replace the weight matrix of linear layers in LSTM and Transformer with PHM.
Finally, they show that these PHM-variants of LSTM and Transformer, match or outperform their vanilla counterparts on a variety of NLP tasks, including NLI, MT, text style transfer, etc.,  while reducing the total number of model parameters.

Overall the paper is quite well written with easy to follow illustrations.
The proposed parametrization seems reasonable, and the empirical validation lends solid credibility to the idea.
I have a couple of questions for the authors:
* What are the practical benefits of this parametrization, particularly in comparison to other ways of reducing parameters, say matrix factorization?
* If hypercomplex spaces only exist in $2^k$-D, is the term hypercomplex justified for arbitrary dimensions?
* Have you analyzed the learned hypercomplex spaces? Can they be interpreted for arbitrary dimensions?

---

> ### Author Response · Authors · 2020-11-21
> **Response to Reviewer1**
>
> Thank you for the positive assessment and insightful comments.
>
> ### On Practical Benefits of Our Proposed Method
> We would like to note that Quaternion representations have already been employed on real-world tasks, such as machine translation (Parcollet et al., 2018a; Tay et al., 2019), while enjoying 75% reduction in the parameter size due to the Hamilton product. Our contribution (PHM) is to generalize such representations and their multiplications; thus, PHM shares practical benefits of Quaternion representations, such as parameter savings, while allowing for architectural flexibility via its customizable hyperparameter $n$. Additionally, Table 2 has also demonstrated that our PHM-Transformer ($n=4$) outperforms Quaternion Transformer on 6 out of 7 machine translation tasks when their parameter savings are at the same level (75% saving). Thus, PHM’s more effective representations than Quaternion representations can also be considered as a practical benefit.
>
> While not within the scope of our paper (our contribution has to satisfy the requirement of generalizing hypercomplex multiplications, not purely reducing parameters), we believe that our paper naturally inspires future research on exploring more practical benefits of our proposed method. If you think that it is necessary, we would like to include more concluding thoughts in the paper to inspire future research.
>
>
> ### On the Term “Hypercomplex”
> For hypercomplex spaces that only exist in $2^k$-D, our method parameterizes and generalizes their multiplications rules. Thus, the term “hypercomplex” has advantages of being self-explanatory for its relations to hypercomplex multiplications, but we are open to suggestions if a better naming exists :)
>
>
> ### On Interpretability of the Learning
> Our experiments on artificial datasets show that PHM can learn to recover complex/Quaternion/Octonion/Sedenion multiplication rules (in the $\mathbf{A}$ matrices). Similarly, even for arbitrary dimensions where such rules are undefined, when we manually specify multiplication rules to generate artificial datasets, PHM can also learn from the dataset to recover such rules: the learned results are interpretable as those manually specified rules.

---

> > ### Comment · AnonReviewer1 · 2020-11-21
> > **Response to Author(s)**
> >
> > Thanks for the detailed response!
> > I'm more convinced about the paper after reading through the responses and checking out the Appendix.
> >
> > I'm going to increase my confidence to 4 but I'll stick with the score of 7.
> > I want to reiterate that the paper is very well written, thorough, and has some very nice easy to follow illustrations.
> > The reason for not increasing the score is that apart from the parameter-saving benefits, I'm not convinced that "hypercomplex multiplications" by themselves are adding any value.

---

> > > ### Author Response · Authors · 2020-11-21
> > > **Thanks!**
> > >
> > > We are glad to hear that you are more convinced! Thanks again for your positive assessment and insightful comments!

---

> > > ### Author Response · Authors · 2020-11-24
> > > **New results for comparing hypercomplex-space and real-space representation learning**
> > >
> > > First and foremost, thank you for the supportive review and extremely constructive comments!
> > >
> > > To show how hypercomplex-space representation learning is useful besides parameter-saving benefits, we have further performed additional experiments on real-space representation learning . In comparison with Table 2 of the paper, the following table adds one row (Tm $n = 4$) for all the seven machine translation tasks, where the vanilla Transformer in real space is at a different scale of parameterization (i.e., −75%).
> > >
> > > | Models   |      #Params      |En-Vi|En-Id|De-En|Ro-En|En-Et|En-Mk|En-Ro|
> > > |:----------|:-------------:|:------:|:------:|:------:|:------:|:------:|:------:|:------:|
> > > | Transformer (Tm)|  44M | 28.43| 47.40| 36.68 |34.60|14.17 |13.96 |22.79|
> > > |Tm $n = 4$| 11M (-75.0%)|    29.23| 33.52| 30.47| 29.39| 11.30| 8.94| 21.11|
> > > |Quaternion Tm| 11M (-75.0%) |28.00 |42.22 |32.83 |30.53 |13.10 |13.67 |18.50|
> > > ||
> > > |PHM-Tm $n = 2$ |22M (-50.0%)   |29.25 |46.32| 35.52| 33.40 |14.98 |13.60 |21.73|
> > > |PHM-Tm $n = 4$ |11M (-75.0%)   |29.13 |44.13| 35.53| 32.74| 14.11| 13.01| 21.19|
> > > |PHM-Tm $n = 8$ |5.5M (-87.5%)  |29.34 |40.81| 34.16| 31.88 |13.08 |12.95| 21.66|
> > > |PHM-Tm $n = 16$ |2.9M (-93.4%)|29.04|33.48 |33.89|31.53| 12.15| 11.97 |19.63|
> > > ||
> > > |PHM-Tm$^†$ $n = 2$ |44M                 |29.54 |49.05| 34.32 |33.88 |14.05 |14.41| 22.18|
> > > |PHM-Tm$^†$  $n = 4$ |22M (-50.0%) |29.17 |46.24| 34.86| 33.80| 14.43 |13.78 |21.91|
> > > |PHM-Tm$^†$  $n = 8$ |11M (-75.0%) |29.47| 43.49 |34.71| 32.59 |13.75 |13.78 |21.43|
> > >
> > > Key findings are:
> > >
> > > * When comparing Quaternion Tm and Tm $n = 4$ side by side, Quaternion Tm outperforms on five out of seven tasks at the same parameterization level.
> > > * When comparing our PHM-Tm $n = 4$ and Tm $n = 4$ side by side, PHM-Tm $n = 4$ outperforms on six out of seven tasks at the same parameterization level.
> > >
> > > Overall, when at the same parameterization level, hypercomplex-space representation learning is more effective than their real-space counterpart. Existing works such as Quaternion CNN (Zhu et al., 14 ECCV’18), Quaternion RNN (Parcollet et al., ICLR’19), and Quaternion Transformer (Tay et al., ACL’19) explained that the weight-sharing property (as illustrated in our Figure 1) in the hypercomplex multiplication makes the Quaternion representation learning more effective.
> > >
> > > Furthermore, in our earlier conversation, we noted that "Table 2 has also demonstrated that our PHM-Transformer ($n=4$) outperforms Quaternion Transformer on 6 out of 7 machine translation tasks when their parameter savings are at the same level (75% saving)". This suggests that PHM further learns more effective representations than the Quaternion representation. To get intuitions for why:
> > >
> > > * As in our responses to Reviewer5 and Reviewer3, we found that PHM can effectively learn predefined real-space operations, such as a rotation in a 3D real space. We have added Section 4 in the revised supplementary materials to include such results.
> > > * As in our responses to Reviewer5, our experiments on artificial datasets also show that PHM can learn to recover complex/Quaternion/Octonion/Sedenion multiplication rules. Specifically, even simple discretization operations can precisely recover the expected geometric properties in predefined hypercomplex spaces. Now we have also added Figure 2 in the revised Section 3.3 to show that PHM layers can empirically learn hypercomplex multiplications.
> > > * As in our earlier conversation, even for arbitrary dimensions where multiplication rules are undefined, when we manually specify multiplication rules to generate artificial datasets, PHM can still learn from the dataset to recover such rules.
> > >
> > > All these results suggest that PHM is both flexible and effective in learning different operations in both real space and hypercomplex space. Intuitively, the operations learned by PHM from data may be more effective than those predefined mathematical rules of hypercomplex spaces for the investigated tasks.
> > >
> > > We hope that the above could clarify on why the representation learning in hypercomplex space may add value to the (probably huge) existing body of literatures on real-space representation learning, since hypercomplex representation for deep learning is still in its infancy. Again, thank you and the other reviewers for helping us make the paper better! We have added such acknowledgements at the end of the revised paper.

---

### Official Review · AnonReviewer5 · 2020-11-06
**Kronecker product as a way to parametrize high-dimensional products**

**Rating:** 8
**Confidence:** 5

**Review:**

This paper builds on the top of standard high-dimensional neural networks (limited to 2,4,8, 16 dimensions) by introducing an elegant way to deal with others dimensions while preserving the internal relation learning capability as well as the reduction of number of parameters. To do so, the linear transformation is turned into a new linear transformation based on the Kronecker product. A sum of Kronecker products is used to "simulate" the different internal relations that could occur in between multi-dimensional components. However, I think that an empirical validation of the ability of the method to recover well-know product, i.e. Hamilton / complex products, is missing.

According to the results, with experiments conducted on 3 different tasks, the proposed approach definitely seems to work and is an important step further to better understand and manipulate high-dimensional algebras with deep neural networks.

Remarks and questions:
1. While the theoretical aspect of the degeneration of the proposed approach to the Hamilton product "sounds" plausible, I would like to see a more empirical demonstration. Can this approach, correctly parametrize a rotation in a 3D space (simple task of learning a single rotation of an object).
2. The quaternion case of this method relies on a set of [-1,0,1] to build the S matrices. This is what ensures the geometrical properties of the quaternion space while doing the different manipulations. This method, however, uses real-valued matrices. It is thus almost certain that the geometrical aspect of the "learnt" algebra is impossible to interpret. Do the authors think that a quantisation of the matrices could help finding pure quaternion / complex / octonions / sedenions matrices ? Such an analysis is crucial to validate the fact that the work proposed here is a generalisation of what have been done before to N dimensions for neural networks.

- Would be great to include the number of parameters in the different Table.

---

> ### Author Response · Authors · 2020-11-21
> **Response to Reviewer5**
>
> Thank you for the positive assessment and insightful comments.
>
> ### On the Suggested Simple Task of Learning a Single Rotation of an Object in a 3D Space
> We performed experiments to learn such a rotation using the PHM layer. Using a 3D rotation matrix $\mathbf{W}$, we created an artificial dataset {($\mathbf{x}_i$, $\mathbf{y}_i$)} where $\mathbf{y}_i$ is generated via 3D rotation of the input: $\mathbf{y}_i = \mathbf{W} \mathbf{x}_i$. As shown in Figure 2 of the revised supplementary materials, the loss converges to zero. This empirical result shows that the PHM layer can learn a single rotation of an object in 3D space. We have added Section 4 in the supplementary materials to include such results.
>
>
> ### On Quantisation for Finding Pure Complex/Quaternion/Octonion/Sedenion Matrices
>
> This is a very smart idea. The deep learning framework requires real-valued parameters to compute gradients during model training. Our experiments on artificial datasets show that PHM can learn to recover complex/Quaternion/Octonion/Sedenion matrices with real values (in the $\mathbf{A}$ matrices) very close to -1, 0, 1. For example, our learned values -1.00000000e+00, -5.18326004e-08, 9.99999881e-01 can be easily rounded to -1, 0, 1, respectively. Therefore, even simple rounding operations can also lead to ideal discretization for PHM to precisely recover hypercomplex multiplication rules.
>
>
> ### On Including the Number of Parameters in Different Tables
> Thank you for the suggestions. We have revised the paper and included the #Params column in Table 1, 2, 4, 5.

---

### Author Response · Authors · 2020-11-21
**Overall Response to All the Reviewers**

We would like to thank all the reviewers for the positive assessment and insightful comments. We have revised our paper based on the comments and provided the individual response to each reviewer.

---

### Comment · ~Abhyuday_Jagannatha4 · 2021-04-12
**releasing implementation**

Hi
could you share the codes please? this would be very helpful and allow others to use your work and build on top of it. thank you

---

> ### Author Response · Authors · 2021-04-13
> **Response to Abhyuday**
>
> Thanks. We are figuring out the process of code release and it may take some time. Recently there have been other third-party implementations that you may consider trying in the meanwhile, such as https://github.com/bayer-science-for-a-better-life/phc-gnn

---

> > ### Comment · ~Abhyuday_Jagannatha4 · 2021-04-15
> > **not usable implementation**
> >
> > Hi
> > thanks for the response,  I tested it but this is very slow and not usable, since in the paper this is written this is of the same time as the original model or faster, could you please tell me and direct me how to have made it faster?  thanks

---

### Comment · ~Rabeeh_Karimi_mahabadi2 · 2021-04-24
**Question on implementation**

Dear Authors,

I tried to implement this paper, but I ran into a lot of numerical issues and I see substantial instability with this method (it gets NaNs often) and so far could not reach the performance required as shown in the paper (for me it gets accuracy which is very below linear layers, even with less division dimension).  Have you applied any normalization or ways to stabilize the method?

Could you share how you initialize the bias and weight matrices please?

thank you

---

### Author Response · Authors · 2021-12-25
**Code release**

Hi everyone, the code is released at [GitHub](https://github.com/astonzhang/Parameterization-of-Hypercomplex-Multiplications). Happy holidays!

---

### Decision · Program_Chairs · 2021-01-07
**Final Decision**

**Decision:**

Accept (Spotlight)

**Comment:**

The authors propose a new parameterization which (across multiple architectures) generalized hypercomplex multiplication and provides for small low dimensions strong performance at substantial parameter savings. All reviewers are happy with the theoretical contributions of the work, but would appreciate additional empirical evidence.